# Wideband Lorenz Resonance Reconfigurable Metasurface for 5G+ Communications

Chun Yang [1] , Chuanchuan Yang [1,*], Cheng Zhang [1], Jiqiang Feng [2,3], Chen Xu [2,3] and Hongbin Li [1,3]

[1] The State Key Laboratory of Advanced Optical Communication Systems and Networks, Peking University, Beijing 100871, China
[2] College of Mathematics and Statistics, Shenzhen University, Shenzhen 518060, China
[3] Peng Cheng Laboratory, Shenzhen 518055, China
* Correspondence: yangchuanchuan@pku.edu.cn

**Abstract:** Reconfigurable intelligent surfaces (RIS) have been considered as a promising solution to enhance the spectrum and energy efficiency of the 5G+ and future 6G wireless communication systems. The performance of RIS will become the key metric of these communication systems. In this paper, we proposed a wideband Lorenz resonance-based metasurface reconfigurable reflectarray (MSRRA) realization scheme with low energy consumption targeted at the center frequency of 28 GHz. A compact voltage bias network for the varactor diodes is carefully designed to reduce losses in RF current and the influence of the bias circuit on the radiating element effectively. An equivalent circuit model for the MSRRA unit cell is also introduced to predict the properties of the MSRRA system, which can be used to optimize the MSRRA design efficiently. In the experimental tests, the proposed MSRRA system can be optimized to cover a dynamic reflection phase range of over $300°$ with a bandwidth of 3.83 GHz, which is consistent with the simulation results. The measured single-scattering beam bandwidth is 1.85 GHz at the center frequency of 28 GHz, which can fully cover the whole n257 channel of 5G NR. The proposed continuous tunable MSRRA can support 5G+ and 6G indoor, short-range links, and outdoor point-to-point communications.

**Keywords:** reconfigurable intelligent surfaces (RIS); reflectarray; equivalent circuit model (EC); metasurface; radar cross section (RCS); scattering beam reconfiguration; transmission line (TL)

## 1. Introduction

RIS-aided wireless communications have recently emerged as a promising solution to enhance the spectrum and energy efficiency of current 5G, 5G+ and future 6G wireless systems [1–8]. Specially, a RIS system allows people to control the electromagnetic wave propagation environment via a set of reconfigurable passive reflecting elements arranged in a certain pattern [9,10]. The appeal of RIS technology is that it consumes less power than the traditional phase-array antennas since it can manipulate incoming electromagnetic wave fronts without employing any power amplifiers or RF chains [11].

Unlike traditional 3D-structured metamaterials, RIS is a 2D artificial structure. By properly designing the phase shift applied by each reflecting element, the thin metal layer on the RIS surface can constructively reform the reflected signal. To achieve this effect, Pancharatnam–Berry phase (PB phase)-based [12–16] and Lorenz resonance-based RIS [17–22] have been introduced continuously. Although PB phase-based RIS are dispersion-free and easy to fabricate, their reconfigurability depends on complex control mechanisms and high energy consumption. Additionally, most of the previous proposals based on reconfigurable PB phase RIS are usually quite complex, relying on the optimization of multiple parameters, which makes them difficult to be applied in real systems [23,24]. Subsequently, RIS based on Lorenz resonances are much easier to achieve reconfigurability with by integrating electronic components or tunable materials into the metal patch on the surface. Current research on Lorenz resonance RIS mainly focus on frequencies up to K–band. Because the electrical size

of the RIS unit cell in the millimeter-wave band is much smaller than the lower-frequency band, it is a great challenge for system design and integration [25–27]. Another system implementation issue that needs to be addressed is the high energy loss of RIS. When the unit cell of the RIS is working at the resonant frequency, it absorbs a large amount of energy of the incident electromagnetic wave and reduces the reflection efficiency of the RIS. In [28], it was reported that the attenuation at the resonant frequency is more than 10 dB and the attenuation difference between the maximum and the minimum exceeds 15 dB. With such a great attenuation loss and fluctuation in the bandwidth, the RIS communication efficiency degrades a lot. Moreover, the reported typical communication bandwidth of the single beam in current RIS based on resonance mechanisms at the millimeter-wave band is hundreds of megahertz [29], which is very limited and cannot meet the high data rate requirements in millimeter-wave communication systems. To overcome the problems addressed above, a wideband MSRRA realization scheme with low energy consumption is proposed as a reflector in this paper. An equivalent circuit model for the MSRRA unit cell is introduced to predict the properties of the MSRRA system. We have designed a compact integrated voltage bias network to support the system's optimization iteratively. The dynamic reflection phase-tuning range of the proposed MSRRA system can reach over $300°$ at a bandwidth of 3.83 GHz in the 28 GHz frequency band. The power consumption of our MSRRA is only about 2.8 W, which can work for about 4 h with a 12 V@1 Ah lithium battery. Scattering beam reconfiguration and a communication bandwidth as high as 1.85 GHz are obtained with the help of the well-designed DC-bias network and the tunable varactors soldered on the surface. With such a wide single-beam bandwidth, the aperture of the MSRRA can be greatly reduced and a significant increase in the data rate of the indoor RIS-aided communication system can be attained.

## 2. Unit Cell Modeling and Design

The proposed equivalent circuit model of the unit cell can set up a direct link among the reflection phases, the bandwidth, and the physical layout of the MSRRA.

### 2.1. DC-Bias Network

A physical RIS with a phase-controlled reflection coefficient is strongly affected by its insertion loss. The losses of the RIS depend on ohmic losses, dielectric losses, and load losses. The last one is the main source of loss fluctuations.

In our proposed system scheme, the DC-bias circuit is properly designed to minimize loss and simplify the structure. The DC-bias point is set at the geometric center of the unit cell, which is along the non-radiating edge of the patch. Thus, the geometry of the metal structure is the same regardless of whether it is seen from an incident or a reflected wave view.

The DC-bias circuit is made up of high-impedance lines, which can greatly reduce RF current loss. A series circuit comprising a resistor and inductor is also designed to choke the RF current. Compared with the RF choke circuit of an open-ended radial line stub with a quarter-wavelength microstrip line, the series circuit of resistor and inductor has a much wider operating frequency range. To avoid the influence due to the self-resonant frequency of the inductor, an inductor whose self-resonant frequency is higher than our operating frequency band should be selected in our design of the DC-bias circuit. So, a thin-film RF/microwave inductor from KYOCERA AVX (L0201068BHSTR) with 0.68 nH inductance and a 31 GHz self-resonant frequency is used in this MSRRA unit cell DC-bias network. To complement the use of the inductor, a high-frequency resistor from VISHAY (FC0603E1001BST1) with 10 K ohm resistance and low parasitic capacitance is also used in the Series DC-bias circuit. These device selections ensure a good isolation in DC and RF fields and reduce additional losses resulting from the biasing network, which usually occur in the reconfigurable RIS elements with an extra DC-bias or controlling network. Moreover, the Series DC-bias circuit composed of resistor and inductor are placed on the bottom metal

layer of the unit cell structure below the ground plate layer; hence, it minimizes the effect of the bias circuit on the scatterer.

### 2.2. Equivalent Circuit Model

According to the propagation path of the incident electromagnetic wave, the equivalent circuit (EC) model of the MSRRA can be modeled by means of using equivalent inductors, capacitors, resistors, and transmission lines, which represent the metal patch, dielectric mediums, the ground plane, and varactor diodes in the MSRRA system. A sketch of the MSRRA and its equivalent circuit are illustrated in Figure 1. The second substrate's medium layer with permittivity $\varepsilon_2$ and propagation constant has no contribution to the MSRRA and is neglected, since it is under the ground plane. The total EC model is a parallel RLC resonant circuit.

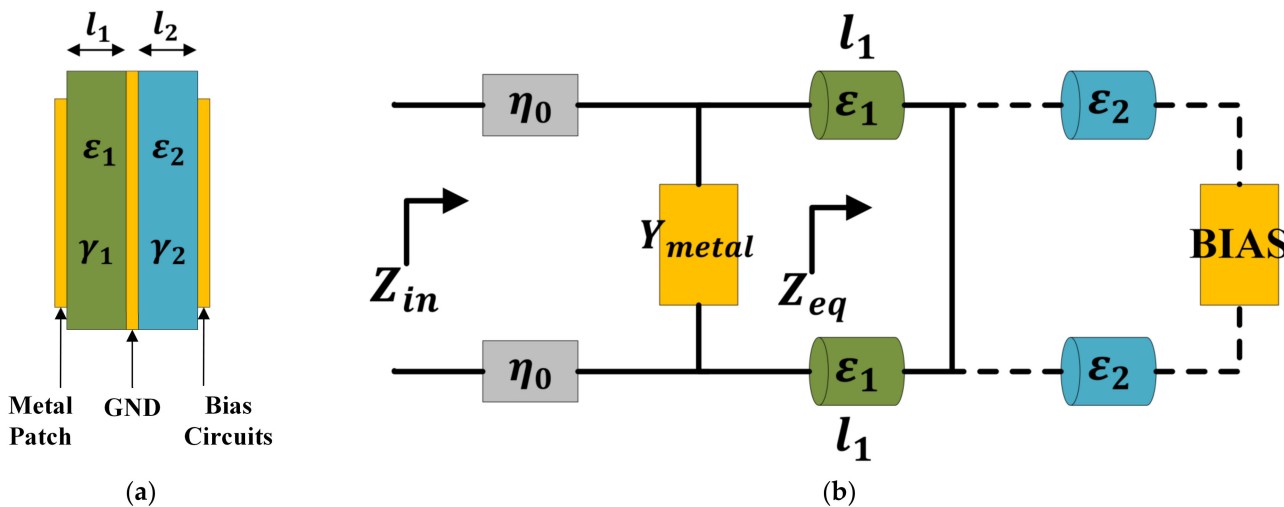

**Figure 1.** (**a**) Side view of the multi-substrate layer MSRRA. Each substrate medium is characterized by its thickness $l_x$, permittivity $\varepsilon_x$, and propagation constant $\gamma_x$. (**b**) Transmission line-based EC model of a multi-substrate layer MSRRA. The metal patch is represented by a shunt admittance.

The analysis of the equivalent circuit model is conducted through the following steps:

(1) Analysis of metal patch with a varactor diode: The metal patch is modeled using lumped R, L, and C components so that a resonant equivalent circuit can be obtained which, together with the varactor, provides equivalent capacitance for the entire EC model.

When the MSRRA is illuminated, the induced electric field appears across the gap between the two microstrip patches. On average, this effect can be equivalent to a sheet capacitance $C_p$. The high-frequency currents flow back and forth between the capacitive gap, so the patches can be equivalented as serials of effective inductive impedance $L_p$ and a resistance $R_p$ as shown in Figure 2a,b. A MACOM varactor diode (MAVR–011020–1411) [30], Ref. [31] is used in this MSRRA, and can provide extra capacitance. With the spice model provided by the manufacturer, we can easily obtain an EC model of the metal patch with a varactor on it, as shown in Figure 2c. Finally, we can set up the whole model, as shown in Figure 2d; the MSRRA can be equivalented to a parallel RLC resonant circuit.

The $C_p$ in Figure 2d is described in Equation (1) [32] as:

$$C_p = \frac{L(\varepsilon_1 + \varepsilon_2)}{\pi} Cosh^{-1}\left(\frac{a}{g}\right), \tag{1}$$

where $L$ is the length of the metal patches, $g$ is the gap width, and $a = 2 \times w + g$ is the total length of the metal patch from edge to edge. The structure is surrounded by $\varepsilon_1$ on one side, and $\varepsilon_2$ on the other. Additionally, closed-form expressions for $L_p$ and $R_p$ in the EC model are not readily available. They must be de-embedded using electromagnetic simulations

of the discontinuity due to their unique geometry. A microstrip patch without a varactor is simulated using full-wave electromagnetic simulation software CST to extract the S-parameters. $L_p$ is calculated using the geometric structure's inherent resonant frequency. Finally, the computation of $R_p$ needs to simulate the reflection coefficient of the patch. Using the measured power in the radiated field, the resistance of the patch can be extracted if the magnitude of the voltage established in the patch slots is known.

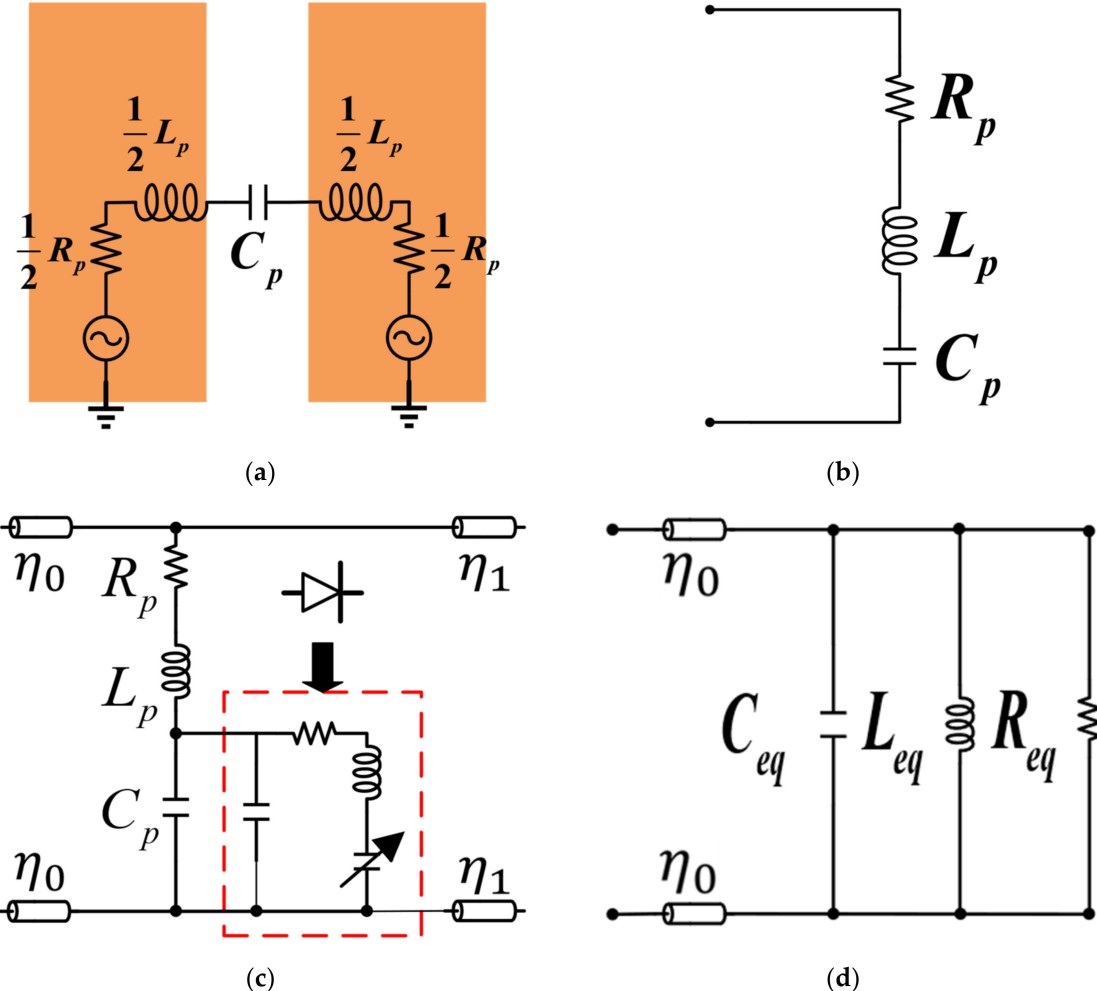

**Figure 2.** Equivalent circuit model of the MSRRA unit cell. (**a**) Metal patch and the equivalent lumped elements. (**b**) EC model of the metal patch. (**c**) EC model of the metal patch with a varactor diode; the varactor diode EC mode is in the red dash square. (**d**) Equivalent circuit model of the MSRRA unit cell.

(2) Analysis of substrate medium and ground plane: Both the substrate medium and ground plane are modeled as transmission lines and are mainly handled as the lumped L component of the entire EC model.

As shown in Figure 1b, $Z_{eq}$ is the input impedance of the substrate with the ground plane and is calculated by:

$$Z_{eq} = j \times Z_{TL} \tan(\beta l), \tag{2}$$

where $l$ is the thickness of the substrate medium, $\beta = 2\pi/\lambda$ is the propagation constant in the medium, $\lambda$ is the wavelength, and $Z_{TL} = \sqrt{\mu/\varepsilon}$ is the impedance introduced by the substrate medium.

The complete EC model of the MSRRA unit cell connects the above two blocks in parallel, as shown in Figure 1b. $Z_{in}$ is the total input impedance and $\Gamma$ is the reflection coefficient, which can be calculated using the following formulas:

$$Z_{in} = Y_{metal} \; // Z_{eq}, \tag{3}$$

$$\Gamma = \left| \frac{\eta_0 - Z_{in}}{\eta_0 + Z_{in}} \right|. \tag{4}$$

In the obtained EC model, the MSRRA unit cell can be regarded as a parallel resonant circuit. The capacitor is introduced by the metal patch (with the varactor) and the inductor is generated by the grounded substrate medium.

Compared with the capacitance introduced by the metal patch and the varactor, the inductance of the metal patch and the varactor is relatively small. Therefore, the patch with the varactor is capacitive. So, if the patch capacitance is designed relatively large, the tuning range of the resonant frequency becomes small within the capacitance variation of the varactor. Conversely, the range of the resonant frequency becomes larger. It is very important to select a reasonable ratio between the patch capacitor and the varactor capacitor.

The grounded substrate can appear capacitive or inductive as the thickness of the medium variates periodically. The periodic thickness is $\lambda/2$. When the thickness is below $\lambda/4$, it is inductive, and it is capacitive when the thickness is greater than $\lambda/4$ and below $\lambda/2$. So, adjusting the thickness of the medium can also adjust the resonant frequency shift within the range where the medium is inductive. On the other hand, since the MSRRA is equivalent to a parallel RLC resonant circuit, the Q factor is proportional to $\sqrt{C/L}$. As the Q factor increases, the slope of the magnitude curve at the resonant point increases while the bandwidth decreases. Conversely, the bandwidth increases. The way to increase the bandwidth is to increase the thickness of the substrate, or reduce the capacitance of the unit cell.

### 2.3. Unit Cell Design

Guided by the EC model of the unit cell, the configuration of the proposed MSRRA unit cell is shown in Figure 3. It is a multi-layer structure; the periodic of the unit cells is 4 mm. On the top layer, two microstrip patches with 0.035 mm thickness make up the unit cell scatterer. The RT5880LZ ($\varepsilon r = 2.2$ and $\tan\delta = 0.0009$) substrate medium with 0.508 mm thickness makes up the second layer of the unit cell. Below the RT5880LZ substrate medium, a metal plane with 0.035 mm thickness acts as the ground plane which helps to improve the reflection efficiency of the MSRRA. The FR-4 ($\varepsilon r = 4.4$ and $\tan\delta = 0.018$) substrate medium with 1.016 mm thickness makes up the second substrate layer, which mainly strengthens the whole unit cell structure. Between the ground plane and FR-4 substrate medium, an FR-4 prepreg layer with 0.2 mm thickness is used to bond the upper and lower structure together, as shown in Figure 3b. The bottom layer of the MSRRA unit cell contains the DC-bias network circuits, which provide DC bias for the varactor diode. All the unit cell structure parameters, which have been iterated and optimized, guided by the EC model, are summarized in Table 1. A varactor is placed between the microstrips patches, adding variable capacitance $C_v$ to the unit cell. The dynamic range of the capacitance $C_v$ is $C_{vmax} = 0.22$ pF to $C_{vmin} = 0.033$ pF for 0–16 V reverse bias voltage.

**Table 1.** Parameters of the proposed MSRRA unit cell.

| Parameters | Value | Parameters | Value |
|:---:|:---:|:---:|:---:|
| P | 4 mm | t | 0.035 mm |
| W | 1 mm | h1 | 0.508 mm |
| L | 2.2 mm | h2 | 0.2 mm |
| g | 0.8 mm | h3 | 1.016 mm |

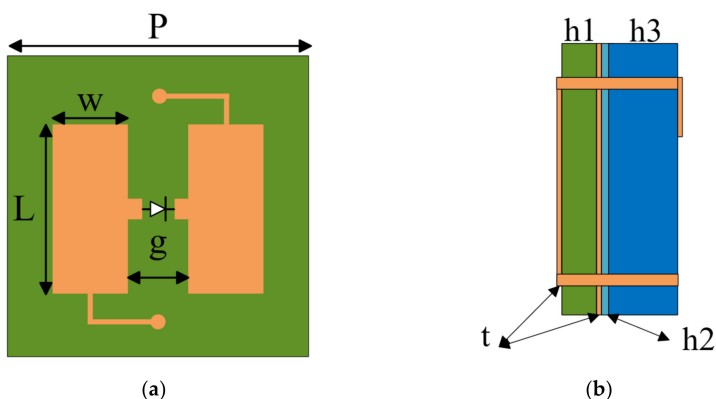

**Figure 3.** Unit cell structure. (**a**) Unit cell top view. (**b**) Unit cell side view.

## 3. Fabrication and Measurement

### 3.1. Fabrication of MSRRA

In order to test the performance and experimentally validate the MSRRA designed in this work, a prototype is fabricated, composed of three independent function bodies based on PCB techniques. The three independent function bodies are inter-connected by the BTB connectors as shown in Figure 4a. The top layer of the MSRRA prototype is the metasurface's aperture, which includes 16 × 16 unit cells and has a square size of 64 mm × 64 mm, as shown in Figure 4b. The middle layer is the bias voltage generator, which has 16 voltage DACs. There are 256 bias voltage lines totally connected with 256 unit cells of the aperture, as shown in Figure 4c. The bottom layer of the prototype is the beam-forming controller, which can independently control one of the 256 bias voltages of the unit cell to realize the free control of the reflected beam. The beam-forming controller is implemented based on an FPGA (Field-Programmable Gate Array), as shown in Figure 4d.

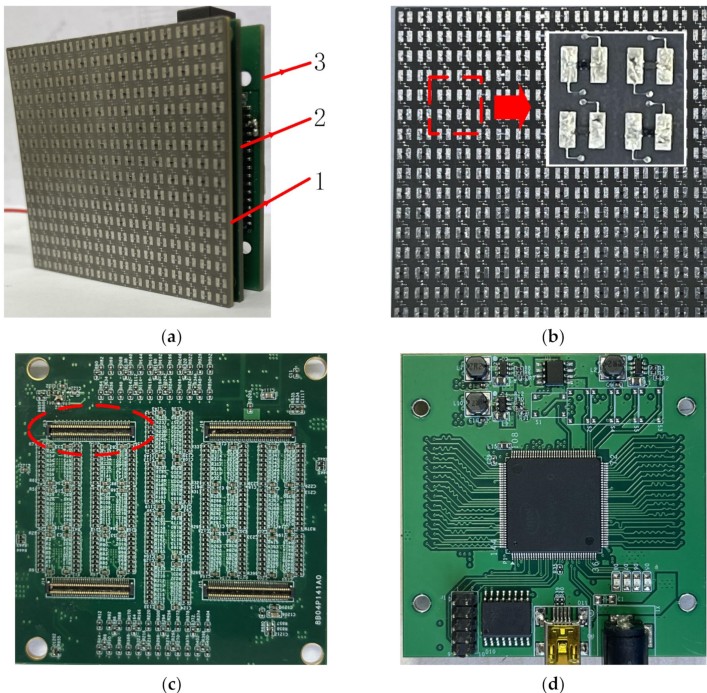

**Figure 4.** MSRRA prototype. (**a**) Laminated structure of the MSRRA prototype, from top to bottom; 1 is the MSRRA reflect aperture; 2 is the bias voltage generator; 3 is the beam-forming controller. (**b**) MSRRA aperture. (**c**) Bias voltage generator; the BTB connector is marked with a red circle; (**d**) Beam-forming controller.

### 3.2. Measurement of the Proposed MSRRA

The MSRRA was measured using a far-field testing system in an anechoic chamber. The block diagram and test scene of the experimental scheme are shown in Figure 5. Two wideband horn antennas, whose working frequency band is from 26 GHz to 40 GHz, are employed as the feed to illuminate the MSRRA aperture and as the receiver to capture the electromagnetic wave reflected from the MSRRA aperture. The two antennas were placed perpendicular to MSRRA, both connected to a Keysight Company model N5525B PNA Network Analyzer (PNA), whose working frequency band is from 10 MHz to 50 GHz.

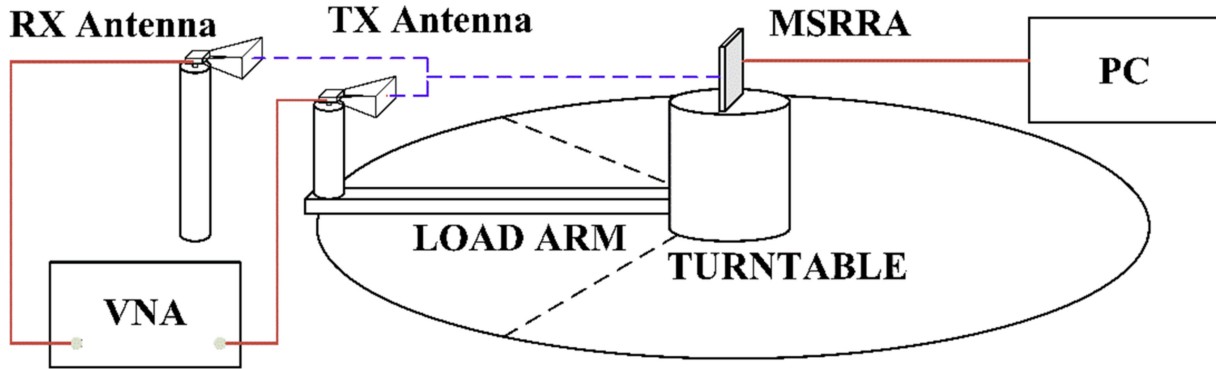

**Figure 5.** Block diagram of experimental scheme.

The aperture diameter of the proposed MSRRA is 64 mm with a corresponding far-field distance of approximately 850 mm. Therefore, the feed horn antenna is placed 900 mm from the center of the turntable and the receiver horn antenna is placed 1000 mm from the center. The receiver horn antenna is mounted at a position higher than the feed horn antenna, and the geometric center of the line connecting the center of the receiver horn antenna and the feed horn antenna is aligned with the phase center of the MMRRA. When the turntable is rotated to 0 degrees, the feed horn antenna will not block the reflected beam from the MSRRA.

The phase shift from the bias voltage of the varactor of the MSRRA was measured, and can reach over 300 degrees from 26.10 GHz to 29.93 GHz, as shown in Figure 6.

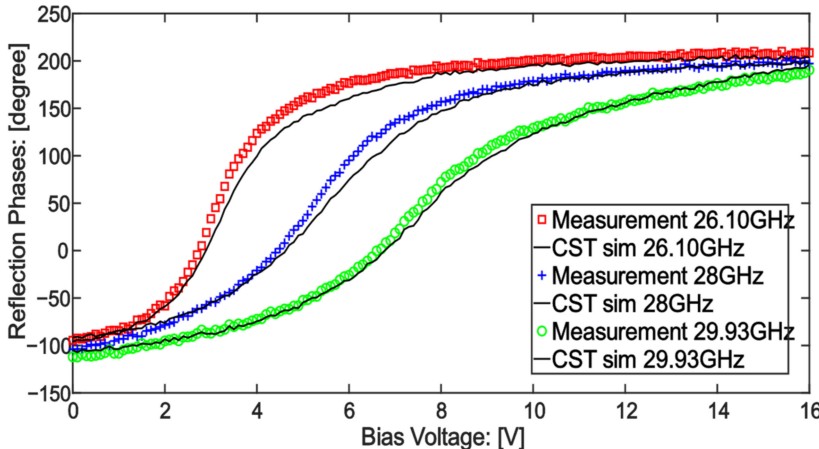

**Figure 6.** MSRRA measurement and simulation results in marked lines and solid lines; they show a good degree of agreement with each other.

The beam-scanning performance at 28.5 GHz of the MSRRA was also measured using the far-field testing system in an anechoic chamber. Some representative measured scan beams with main beam directions from −45° to 45°, stepped by 1°, are shown in Figure 7.

Owning to the well-performing MSRRA unit cell and the effectively optimized phase distribution, well-defined scan beams are successfully obtained in the prototype.

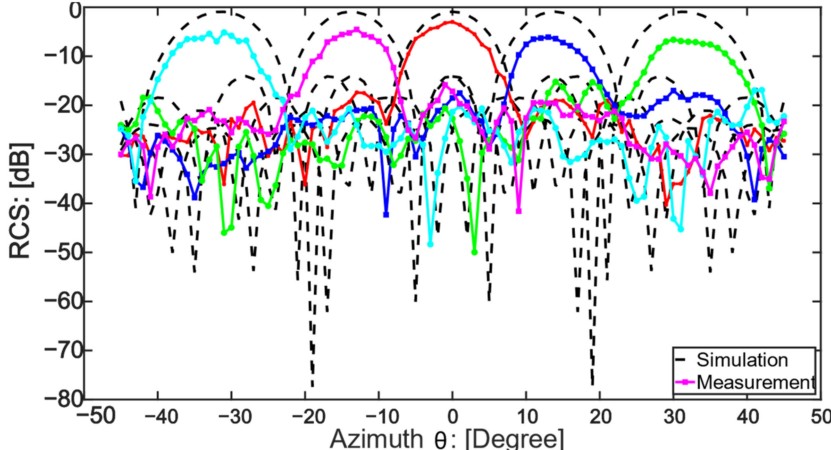

**Figure 7.** Far-field measurement and simulation results for several steering beams at 28.5 GHz. They show a good degree of agreement with each other. The direction of the reflect single beam is −33° (light blue line), −13° (pink line), 0° (red line), 13° (blue line) and 33° (green line).

The bandwidth performance of the MSRRA phototype was also measured. As shown in Figure 8, it is clear that, within −3 dB gain, and within the single beam direction, the beam from 27.05 GHz to 28.9 GHz remains very close; the bandwidth of the single beam is 1.85 GHz.

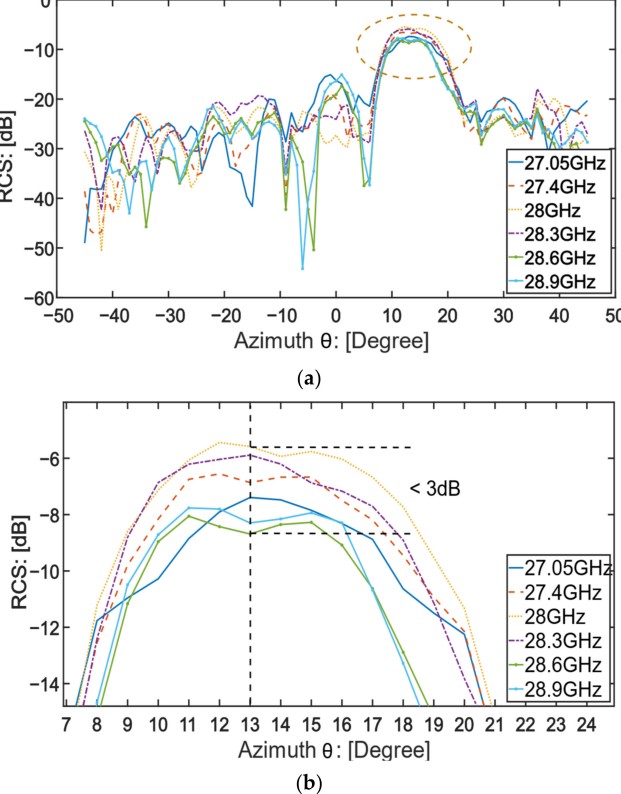

**Figure 8.** Bandwidth of single beam within −3 dB gain. (**a**) From 27.05 GHz to 28.9 GHz, the main beam remains very close. (**b**) Zoom in on the dashed circle of (**a**).

## 4. Conclusions

In this paper, a wideband MSRRA realization with low energy consumption is proposed targeted at the center frequency of 28 GHz. We have designed a circuit for providing DC-bias voltage for the varactors, which can greatly reduce the loss of the RF current and the influence of the bias circuit on the radiating element. Additionally, an EC model for an MSRRA unit cell is also introduced. Guided by the EC model, we designed a simple unit cell structure suitable for operating at a millimeter-wave frequency band. The proposed MSRRA system can cover a dynamic reflection phase range of over $300°$ from 26.10 GHz to 29.93 GHz. The single-beam bandwidth performance of the proposed MSRRA phototype is also measured as 1.85 GHz. The proposed MSRRA can fully cover the whole n257 channel of 5G NR. For future 5G+ and 6G applications, the proposed MSRRA prototype has a broad application prospect.

**Author Contributions:** C.Y. (Chun Yang) designed the structure of the MSRRA, proposed the EC model for the unit cell, designed the experiments, analyzed the experimental data, and wrote the paper; C.Y. (Chuanchuan Yang) examined and approved the structure of the MSRRA, the EC model, and the paper. C.Z., J.F., C.X. and H.L. analyzed the experimental data and approved the paper. All authors have read and agreed to the published version of the manuscript.

**Funding:** This research was funded by the National Key R&D Program of China under Grant 2020YFB1806405 and the Peng Cheng Laboratory under Grant PCL2021A04.

**Institutional Review Board Statement:** Not applicable.

**Informed Consent Statement:** Not applicable.

**Data Availability Statement:** Not applicable.

**Conflicts of Interest:** The authors declare no conflict of interest.

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
