# Peer review of "Wideband Lorenz Resonance Reconfigurable Metasurface for 5G+ Communications"

_electronics, doi:10.3390/electronics11244105_

Round 1

Reviewer 1 Report

This paper is too much interesting and organized very well, I would like to recommend it for publishing in this journal only I have 2 simple comments for authors 

1) don't use solid lines for all diagram in a Figure such as Fig.10 

2) the main antenna under the metasurface is not clear please present it very well 

Reviewer 2 Report

1- The language of the paper requires lot of modifications. 

2- he authors claim that the obtained bandwidth was not reported before. However, they did not compare the relative bandwidth to other published results. Actually, the obtained relative bandwidth is not larger than other previous studies.

3- Ref. [30] is not the appropriate  reference for Eq. (1). Eq. (1) is not mentioned in this reference.

4- In page 6, line 183, the authors mentioned that the quality factor is proportional to the sqrt(C/L). This is not correct.

5- The analysis is in Sec. 2.2 is well know in the literature. Moreover, it is not presented in good way.

Round 2

Reviewer 2 Report

The authors replied on all commnets.